# Avian Pathogenic *Escherichia coli* (APEC) in Broiler Breeders: An Overview

**DOI:** 10.3390/pathogens12111280

**Published:** 2023-10-26

**Authors:** Jiddu Joseph, Li Zhang, Pratima Adhikari, Jeffrey D. Evans, Reshma Ramachandran

**Affiliations:** 1Department of Poultry Science, Mississippi State University, Mississippi State, MS 39762, USA; jj2556@msstate.edu (J.J.); lz245@msstate.edu (L.Z.); pratima.adhikari@msstate.edu (P.A.); 2Poultry Research Unit, Agriculture Research Service, United States Department of Agriculture (USDA), Mississippi State, MS 39762, USA; jeff.evans@usda.gov

**Keywords:** avian pathogenic *E. coli*, broiler breeders, colibacillosis, vertical transmission, first-week mortality

## Abstract

Poultry meat is one of the major animal protein sources necessary to meet the global protein demand. Sustainability in broiler production is the key to achieving its continuous supply, and broiler breeders play a critical role in maintaining this sustainability by providing good quality chicks. Colibacillosis, the disease caused by avian pathogenic *Escherichia coli* (APEC), causes severe economic losses to the poultry industry globally. Moreover, APEC causes an additional burden among broiler breeders, such as a decrease in egg production and mortality among these birds. There is vertical transmission of APEC to the broiler chicks through eggs, resulting in increased first-week mortality and subsequent horizontal transmission at the hatchery. In this regard, the vertical transmission of antibiotic resistance genes is another concern that needs attention. Controlling several diseases in broiler breeders would possibly reduce the first-week mortality in chicks, thereby maintaining the production level. For that, constant monitoring of the bacterial populations is critical. Moreover, amidst the increased antibiotic resistance pattern, more focus on alternative treatment strategies like vaccines, probiotics, and bacteriophages is necessary. Future research focusing on strategies to mitigate APEC in broiler breeders would be one of the finest solutions for sustainable broiler production.

## 1. Introduction

The United States is a major contributor to the world’s broiler supply. It has the highest broiler production with around 20 million metric tons of chicken meat production annually [1]. The value of broiler production from the U.S. in 2022 was $50.4 billion, 60% higher than in 2021 [2]. As the demand for broiler meat increases, there is a need to improve production [1]. However, with the wide acceptance of the No Antibiotic Ever (NAE) system across the U.S., disease control among the poultry population has become a major challenge. Meat- and poultry-related illnesses are causing an economic loss of around $20.3 billion annually. Therefore, it is critical to develop effective control measures against bacterial diseases, such as colibacillosis, salmonellosis, and campylobacteriosis, which are causing a major loss to the industry [3]. Additionally, maintaining a continuous supply of broilers in the market requires an excellent strategy for managing broilers and broiler breeders [4].

Broiler breeders are the parents of broilers and play a critical role in developing a healthy broiler flock [5]. Decreased fertility, hatchability, and egg production are some of the major challenges faced by the broiler breeder industry globally [6]. These challenges could be due to a combination of factors, such as improper management, stress, inadequate nutrition, immunosuppression, and exposure to disease agents. Identifying these factors and finding timely solutions are the key strategies for producing healthy broilers for the market [7]. 

*Escherichia coli* (*E. coli*) is a bacterium commonly found in the normal intestinal flora of humans, other mammals, animals, and birds. However, some of the strains can act as major pathogens causing severe disease and high death tolls [8]. Extraintestinal pathogenic *E. coli* (ExPEC) causes the disease called colibacillosis in animals and birds. ExPEC has divisions, such as uropathogenic *E. coli*, newborn meningitic *E. coli*, septicemia-associated *E. coli*, and avian pathogenic *E. coli* (APEC). The APEC causes disease specifically in birds, including chickens, ducks, and turkeys [9]. Avian colibacillosis is manifested in diverse ways, such as peritonitis, salpingitis, yolk sac infection, and cellulitis, which are localized infections while colisepticemia, pericarditis, airsacculitis, coligranuloma, and arthritis are systemic infections leading to high morbidity and mortality [10,11]. This is one of the most commonly occurring and economically devastating bacterial diseases among poultry worldwide [11].

The primary routes of APEC entry include fecal–oral, respiratory, and vaginal origin (ascending route) through the cloaca [9]. The fecal–oral and respiratory-type colibacillosis has been studied the most [12]. However, the ascending route through the cloaca is also very important as it results in outbreaks characterized by salpingitis–peritonitis syndrome in broiler breeders and causes huge economic losses due to decreased egg production and quality, increased mortality, and the cost associated with treatment, culling of birds, and disposal of carcasses. This is further exacerbated by the vertical transmission of APEC from breeders to progenies leading to yolk sac infection, omphalitis, and increased first-week mortality among broiler chicks [13,14]. Moreover, infected chicks, dead embryos, and contaminated eggshells act as potential sources of infection in the hatchery leading to horizontal transmission among uninfected chicks and contamination of equipment [15,16]. Additionally, vertical transmission facilitates the transfer of antibiotic-resistant APEC strains down the production cycle, making their control extremely difficult [17,18]. 

Traditionally, prevention and control of APEC mainly focus on hygiene achieved through strict biosecurity measures and proper cleaning of facilities using disinfectants and antimicrobials, treatment using antibiotics, and vaccination strategies in birds [19].Most of these traditional methods become ineffective over time mainly because of the evolution of resistant bacterial species [20]. The emergence of antibiotic-resistant bacterial species has led to restrictions on antibiotic use such as for treatment purposes [21,22,23]. As a result, alternatives to antibiotics, such as probiotics, prebiotics, and phage therapy, are gaining more acceptance [24]. In the case of vaccines, different types, such as inactivated, subunit, and live attenuated vaccines, are available in the market for APEC prevention, such as Poulvac *E. coli* and Nobilis *E. coli*, but they provide no or limited cross-protection between strains because of the continuous evolution of the bacterial strains. Hence, more potent vaccines targeting heterogenous APEC strains need to be developed [12]. In addition, for the effective control of APEC among broilers, it should be first controlled at the highest level of the production pyramid (Figure 1), i.e., among broiler breeders, and subsequently, it should be maintained down to the lowest level [24].

This review discusses the importance of APEC in broiler breeders as they have a critical role in APEC transmission and control in broiler production. We reviewed the broiler breeder industry in general, the factors contributing towards APEC infection in breeders and transmission to broilers, strategies needed to prevent and control the disease, and future research directions.

## 2. Broiler Breeder Industry and Its Challenges 

The necessity for animal protein, especially poultry meat, is increasing day by day throughout the world. Per capita meat consumption in the U.S. in 2021 shows that the most consumed meat type is broiler meat, which is about 95.6 pounds. However, there was a decrease in per capita consumption in 2022, but forecasts predict that it will again increase to 101.6 pounds in 2031 [25,26]. The high preference for broiler meat is mainly because of its availability, low-fat content, and absence of religious taboos [27]. To meet the increasing demand, there should be a continuous supply of broilers to the market. To achieve this, there is a need to effectively select and manage (health and nutrition) the broiler parent flock, the broiler breeders [27].

In the 1980s and 1990s, the United States and Europe became the major markets for broiler meat. As the market matured, primary emphasis was given to breast muscle yield, and it became the main selection criterion [27]. The three primary breeding companies in the world that have access to all the genetic stock are Cobb-Vantress (Cobb 500, Cobb 700, and Cobb MV male), Aviagen (Ross, Arbor Acres, and Lohmann Indian River), and Hubbard [28]. These primary breeder companies have the pure line elite (pedigree), great-grandparent, and grandparent flocks, which are less in number at the top of the production pyramid. The commercial broiler companies own the parent broiler breeder and broiler flocks, which are large in numbers at the bottom of the pyramid (Figure 1) [29]. 

From the early 1990s to the 2000s, due to intense genetic selection and improvement in management practices, there was a significant improvement in production performances, such as live weight (2.67 kg from 1 kg) and feed efficiency (1.63 kg feed/kg bodyweight from 4.7 kg feed/kg body weight), and a decrease in mortality (3.6% from 18%) as well as market age (6 weeks from 16 weeks) [30]. However, this intense genetic selection for rapid growth has led to poor reproductive efficiency in broiler breeders causing decreased fertility, hatchability, and egg production [28]. Also, metabolic and physiological disorders such as ascites, tibial dyschondroplasia, and twisted leg syndrome have become more common [29,30,31]. 

With poultry production moving towards the No Antibiotics Ever (NAE) system, another major challenge faced by the broiler breeder industry is the difficulty in controlling bacterial, viral, and parasitic diseases among birds [21]. The primary broiler breeder companies follow strict biosecurity; however, the integrators with broiler parents face challenges in maintaining strict biosecurity to prevent the occurrence of diseases as these birds are kept for a longer period of time compared to broilers [32]. To maintain flock fertility, spike males are introduced to older flocks, and this may represent a biosecurity breach as there is a high chance of pathogen transmission [33]. Moreover, non-integrator farms are more difficult to control as they are not under the direct supervision of the parent company [32]. Broiler breeders represent the point where the pathogens capable of vertical transmission are introduced to the production pyramid, causing further horizontal transmission among the progeny. *Mycoplasma gallisepticum*, *Salmonella* Enteritidis, *E. coli*, and Avian encephalomyelitis virus are some pathogens that are known to vertically transmit in poultry. *Mycoplasma* and *Salmonella* are pathogens of food safety concern to humans, and hence, they are closely monitored under the National Poultry Improvement Plan program standards (NPIP) to prevent vertical transmission. However, in the case of APEC, there is no direct evidence of zoonotic potential [34]. However, various studies have identified close genomic similarities between certain human and avian isolates [35,36,37]. This in fact triggers the need to closely monitor the human transmission potential of APEC isolates. Furthermore, eggs from different breeder flocks are hatched together and if the chicks have a pathogen that is vertically transmitted, it may horizontally transmit in the hatchery, and above that, the transportation to different places may exacerbate the chances of spread [24]. Chicks coming from different breeder flocks kept on a single farm are also a potential source of infection [38]. Another challenge is the limited vaccination options among broiler breeders (booster doses) and the lack of cross-protection among commercially available vaccines against different strains [12,39,40]. Identifying new vaccine targets or developing vaccines that provide cross-protection may help to limit the disease in broiler breeders. More about vaccines are further discussed in the following sections.

## 3. APEC in Broiler Breeders

The economic impact of APEC infection in poultry worldwide is extremely high, causing severe losses due to decreased production, the cost of carcass disposal, and the increased cost of medications [41]. In the Netherlands, estimates showed a loss of € 3.7 million due to APEC infections in poultry farms per harvest [42]. In Indonesia, the estimated loss was 1049 million and 992 million U.S. dollars per harvest for broilers and layers, respectively [43]. In the U.S., it has been estimated that economic losses to the broiler industry can be as high as 40 million dollars annually due to carcass condemnation alone [44]. However, the data regarding the overall economic impact of APEC on the U.S. poultry industry are not available. The referenced estimates may be considered conservative as they omitted APEC-associated detrimental impacts on broiler breeders [45,46,47]. Hence, there is a need to estimate the overall economic impact caused by APEC on U.S. poultry production.

### 3.1. Broiler Breeder Hens

Among several manifestations of colibacillosis in chickens, salpingitis–peritonitis syndrome has received relatively little attention [48]. The hens get the infection mainly through three routes: (1) ascending infection through the cloaca or translocation of bacteria from the (2) respiratory tract or (3) intestinal lumen. Among these, the most important routes of infection are the respiratory and fecal–oral routes; however, the ascending infection through the cloaca is also critical for this infection [48,49]. Furthermore, among multiple factors causing mortality in broiler breeder hens, the highest mortality was due to salpingitis–peritonitis syndrome, and *E. coli* was one of the major pathogens causing this systemic infection in broiler breeders [15]. Initially, the infection in the reproductive tract will be asymptomatic and later proceed to severe septicemia and death. Peritonitis is usually found in acute cases mostly as a complication of oviduct infection [50]. Giovanardi et al. [45] reported vertical transmission of APEC from broiler breeders to their progeny in an integrated poultry production chain and suggested chronic salpingitis in hens as a potential risk factor for transmitting APEC through eggs to progenies. Studies show that APEC infection and first-week mortality among chicks from aged broiler breeders (>50 weeks) are higher than from young (<30 weeks) birds, and the plausible reasons are a decrease in the eggshell quality with age and immunosuppression in birds after the peak production period [16,51]. However, there is only limited information on the salpingitis–peritonitis syndrome in broiler breeders; hence, further studies focused on the pathogenesis of APEC isolates causing salpingitis, as well as factors promoting salpingitis in broiler breeders. This research needs to be carried out to develop effective control strategies.

### 3.2. Broiler Breeder Males

Male fertility in broiler breeders is another important aspect of production. Previous reports show that intestinal bacterial populations can invade the male reproductive system as the cloaca is the common opening for reproductive and digestive systems, thus, affecting sperm motility and fertility [52]. Bacterial orchitis may occur due to *E. coli* infection of the broiler breeder’s male reproductive system. *E. coli* mainly invade through the hematogenous route due to septicemia and affect the tubular architecture [53]. The presence of pathogenic bacteria in the semen has been previously reported [54]. Furthermore, the motility of broiler breeder semen samples in the presence of pathogenic bacteria such as *E. coli*, *Salmonella*, *Campylobacter*, *Clostridium*, *Bifidobacterium*, and *Lactobacillus* has been analyzed, and the presence of *E. coli* in semen affected its motility as well as decreased the pH of the semen sample [55]. Additionally, other reports also validate that the presence of *E. coli* in the semen could affect sperm motility [56]. This implies that fertility could be affected by APEC infection in males due to low sperm motility and thus, lead to decreased chick production. In the broiler breeder industry, even though a large number of females are fertilized by a single male (1 male:10 females), there is no monitoring of the semen quality of roosters. Furthermore, when the male reproductive tract gets infected, there is a high chance of transmitting *E. coli* to hens through copulation [56,57,58]. In a study conducted by USDA’s National Animal Health Monitoring System (NAHMS), the introduction of spike males (superior young males) to older breeder flocks to enhance fertility was found to increase *E. coli* peritonitis as it is a breach in biosecurity [59]. To the best of our knowledge, there are no estimates of mortality due to APEC infections in broiler breeder males; however, periodic monitoring of roosters for *E. coli* infections will help to control the disease transmission to hens as well as throughout the broiler production chain.

### 3.3. Virulence Characteristics

Regular characterization of bacterial populations based on their virulence properties is necessary to get updated knowledge about the virulence patterns and to develop effective control measures. Multiple virulence-associated genes (VAGs) that enable *E. coli* to attach, invade, colonize, replicate, and damage the host cells, as well as evade the host immune response, have been studied. However, there are only limited studies on the properties of APEC isolates specifically collected from broiler breeders. The study on the virulence-associated genes of 28 clinical isolates from broiler breeders in the U.S. using conventional PCR reported *iroN* (iron acquisition), *iss* and *ompT* (protectins), and *hlyF* (toxin production) as the most prevalent genes (78.6%) among the isolates [20]. Studies from Thailand and Australia with 200 clinical samples from broilers and broiler breeders using pentaplex PCR showed a high prevalence (100%) for *ompT* and *hlyF* [60]. About 256 broiler breeder APEC non-clinical isolates from Korea were examined using conventional PCR and showed *ompT* as the highly prevalent virulence gene (26%) [61]. Moreover, clinical isolates from broiler breeders in Canada also showed another pattern of prevalent genes among the isolates using conventional PCR (*cvaC-etsB-fyuA-ireA-iroN-iss-iutA-ompT-sitA*) with *sitA* (iron acquisition) as the most prevalent gene (92.75%) [62]. A study conducted in Brazil showed that virulence factors such as type-1 fimbriae and pili type-1 encoded by many genes of the series *fim* aid in their attachment to the reproductive tract of breeders [51]. Similar results were observed in the studies conducted in different parts of the world for various isolates from peritonitis/salpingitis lesions [63,64,65]. Johnson et al. [66] identified five virulence-associated genes that can predict APEC primarily isolated from broilers (*iroN*, *iss*, *iutA*, *hlyF*, and *ompT*) using pentaplex PCR; however, no specific predictor or marker genes have been identified in broiler breeders [10,20,42]. Therefore, studies may focus on identifying conservative genes/virulence factors using advanced technologies like high-throughput sequencing that can reveal APEC so that disease diagnosis becomes easier and more reliable using some multiplex PCR methods [67,68,69]. 

Phylogenetic classification of broiler breeder isolates shows the prevalence of B2 phylogroup (49%), and the identified prevalence agrees with the general pattern in poultry which followed the old Clermont classification which included phylogroups A, B1, B2, and D [70]. However, according to the new Clermont classification which includes phylogroups A, B1, B, C, D, E, F, and clade 1, B2 is still a major phylogroup but other phylogroups have shown some shifts in some categories such as A to C and D to E or F as mentioned before, which should be considered in future studies while classifying broiler breeder APEC isolates [71,72,73]. Also, previous whole genome sequencing data showed some prevalent sequence types such as ST95, ST117, and ST131 from broiler breeder APEC isolates in which ST95 represented most isolates which caused salpingitis and vertical transmission. Further studies in this area are required to provide a comprehensive idea about the sequence types among broiler breeder APEC isolates [16,71,72]. Details regarding the genotypic virulence characterization of broiler breeder APEC isolates are provided in Table 1.

Phenotypic virulence characterization of isolates is equally important as genotypic virulence characterization. Understanding the phenotypic variation relative to genetic differences is critical to understanding bacterial characteristics and helps to identify specific factors responsible for bacterial virulence and pathogenesis [42]. Day-old chick challenge and embryo lethality assays are some convenient methods to phenotypically characterize the isolates based on their virulence potential [75,76]. Traditionally, these assays use APEC clinical and non-clinical isolates and categorize them into highly virulent, intermediate virulent, or avirulent based on the percentage of embryo/chick mortality and lesions produced. Additionally, the relationship between genotypic and phenotypic virulence can also be identified [75]. Oh et al. [74] phenotypically characterized clinical APEC isolates from broiler breeder, broiler, and native chicken of Korea. They found that, as the number of virulence-associated genes increased per isolate, there was an increase in phenotypic virulence per isolate (r = 0.60, *p* < 0.05). Similar results were observed during phenotypic virulence characterization using clinical broiler breeder APEC isolates [77]. However, no such relationship was observed during phenotypic virulence characterization using broiler clinical APEC isolates [75].

In vitro cell culture studies are also helpful to characterize the isolates based on their adherence and invasive properties to eukaryotic cells. As these in vitro studies can mimic the host–pathogen interaction, the adhesion and invasion potential can be used to estimate its relationship with genotypic and phenotypic virulence factors of the bacterial isolates. Moreover, the above-mentioned data can be used to estimate the specific factors responsible for virulence based on a whole genome aspect and to develop effective control methods [78]. Even though the genotypic and phenotypic data provide information about the characteristics of broiler breeder APEC isolates, the vast genetic diversity is a challenge for its control. Continuous monitoring of the APEC isolates and characterizing them both phenotypically and genotypically are critical because of the evolution of bacteria over time. 

### 3.4. Antibiotic Resistance Characteristics

One of the major concerns regarding APEC colonization in broiler breeders is the vertical transmission of antibiotic-resistant genes down the poultry production pyramid [79]. Additionally, the possibility of the zoonotic potential of APEC, including the transfer of antibiotic-resistant genes to humans, has been suggested by Johnson et al. [66]. Therefore, it has been recommended that antibiotics utilized in animal production solely be used for the therapeutic treatment of disease and should utilize antibiotics less commonly applied to human medicine. As the antimicrobial resistance pattern among poultry is continuously evolving, it is important to closely monitor the resistance pattern of isolates from breeders [80]. A recent study on the antimicrobial resistance pattern among the *E. coli* isolates obtained from the ovaries of broiler breeders was thought-provoking, as high resistance patterns toward various first-line antibiotics were observed [17]. As reports showed the vertical transmission of various antibiotic-resistant genes, identifying *E. coli* isolates in the ovary with similar resistance patterns is quite disturbing. Another major challenge faced by the poultry industry is the prevalence of multi-drug resistance among the APEC isolates. There is evidence of recovery of multi-drug-resistant *E. coli* from the reproductive tract and egg contents of broiler breeders (64%) [17]. Joseph et al. [20] identified about 10.7% of the clinical broiler breeder APEC isolates as multi-drug-resistant, which was a significant finding during the study. Most of the studies mentioned above used conventional PCR methods to identify the resistance genes. Even though strict biosecurity is practiced at the top level of the breeding pyramid, there are still challenges at the lower levels, especially the broiler parents. Hence, the proper use of antibiotics in broiler breeders is critical [79,80]. Furthermore, more focus on alternative treatment approaches may also help to prevent antibiotic resistance. Currently, the poultry industry is more focused on the NAE production system, and it generates more than 50% of birds produced in the U.S. [34]. Details regarding antibiotic resistance patterns among isolates from broiler breeders and first-week broiler chicks are demonstrated in Table 2.

### 3.5. Metal Resistance Characteristics

Another important challenge besides antibiotic resistance among APEC isolates is the resistance towards heavy metals, which were used as growth promoters in the past, and quaternary ammonium compounds, which are part of disinfectants used for cleaning poultry facilities [86]. Broiler and broiler breeder APEC isolates were reported to show a high prevalence of the *arsC* gene which codes for arsenic resistance. The use of arsenic-containing feed additive, Roxarsone, in poultry might be a possible reason for this increased prevalence [20,87]. Furthermore, the presence of more than 90% silver resistance among broiler breeder APEC isolates from the U.S. was alarming because silver compounds were used as feed additives [20]. Other metal resistance genes that code for tellurite and mercury are also prevalent among broiler breeder, turkey, and broiler APEC isolates [87,88,89]. Copper resistance genes (20%) were present among the broiler breeder clinical APEC isolates tested in the U.S. However, phenotypically all isolates were susceptible to copper. This calls for continuous monitoring of their vertical and horizontal transmission. Furthermore, the inappropriate use of disinfectants such as the continuous use of subinhibitory concentrations in poultry facilities might be a reason for the presence of resistance against quaternary ammonium compounds among APEC isolates [20]. New alternative strategies for disinfection or proper directives to use disinfectants at the proper concentrations for cleaning should be strictly followed to prevent bacteria from developing resistance.

## 4. APEC Transmission from Broiler Breeders

### 4.1. Vertical Transmission

Transmission of bacteria from broiler breeders to their progeny, possibly due to colonization in the reproductive tract or due to penetration of eggshell, is described as vertical transmission. It is a major concern for APEC transmission and spread [13]. However, only recently the high degree of vertical transfer of APEC from breeders to broiler chicks has been identified. Giovanardi et al. [44] identified that the strains of *E. coli* isolated from broiler chicks were similar to those found from their parents. They identified the presence of O78 (36.3%) and O136 (26%) serogroups among both parent flocks and chicks along with some virulence-associated genes such as *fim/tsh/iuc* that helped to define pathotypes. Furthermore, multi-locus sequence typing (MLST) of isolates from broiler breeders, newly hatched chicks, and dead broiler chicks during the first week using pulse field gel electrophoresis (PFGE) showed the possibility of vertical transmission [15,16]. Similarly, the same clones of APEC isolates that caused salpingitis–peritonitis syndrome were recovered from the hatchery and day-old broiler chicks, again showing the possibility of vertical and horizontal transmission [16,46]. 

The vertical transmission is further confirmed by various other studies that reported the transmission of antibiotic resistance genes of APEC from broiler parents to chicks. The strain obtained from the chicks that died in the first week was also recovered from the broiler parents by following fluoroquinolone-resistant *E. coli* that was vertically transmitted down the integrated broiler production chain [82]. Furthermore, the extended-spectrum beta-lactamase (ESBL) and plasmid ampicillinase C (*pAmpC*) resistance were also found to be vertically transmitted from broiler breeders to broilers [18,79,83]. Another relevant finding supporting the vertical transmission potential is the recovery of tetracycline-resistant *E. coli* from the ovary and egg contents of broiler breeders [84]. All these findings strongly support the possibility of vertical transmission (Table 3), but above that, the increasing antibiotic resistance pattern in *E. coli* should be strictly monitored even though the use of antibiotics is highly controlled. In addition, developing a good strategy to control APEC infection in broilers should primarily focus on the vertical transmission potential of this pathogen as this is an underestimated factor that could cause severe loss.

One of the critical factors affecting vertical transmission is the age of the breeder flock. Monroy et al. [51] reported that the age of broiler breeders affects the colonizing ability of APEC in the host tissue by studying the in vitro adherence ability of *E. coli* in the ciliated oviduct epithelium. Interestingly, the adherence was more towards the oviduct from aged breeders (50 weeks old) than from young (12 weeks old) breeders. This difference in the adherence pattern might be due to differences in the oviduct epithelium cell types in young and aged hens, which in turn might be the effect of hormones and egg production status. Thus, ciliated epithelium found in egg-laying hens oviducts promotes the colonization of *E. coli* [75]. Moreover, the egg albumen secreted by the reproductive tract during egg production is assumed to further promote *E. coli* attachment. These host factors along with *E. coli* fimbrial proteins contribute towards oviduct adherence and colonization [75]. Another contributing factor is the decrease in eggshell thickness with age. One study showed increased first-week mortality due to APEC infection in chicks that hatched from older parents (1.3%) (>60 weeks) with thin-shelled eggs as opposed to younger parents (0.5%) (>30 weeks) with thick-shelled eggs. Also, floor eggs from older parents were found to have higher first-week mortality compared to that from the younger flock [16]. Therefore, constant monitoring of the flocks with an emphasis on age is suggested to limit vertical transmission and APEC infections in broiler chicks. Figure 2 demonstrates the routes and mechanisms for vertical and horizontal transmissions of APEC in poultry.

### 4.2. Horizontal Transmission

In horizontal transmission, the bacteria from the infected birds in a flock get transmitted to the non-infected ones through contaminated body fluids, aerosols, feathers, and feces. Broiler breeders can transmit *E. coli* among themselves and vertically to their progenies, and the vertically transmitted APEC can increase the first-week mortality among the chicks because of the horizontal transmission happening at the hatchery and farm [80,82]. In the hatchery, the eggs infected from the hens’ oviduct could potentially transmit APEC to other embryos during incubation, as well as in the hatching basket, non-infected chicks can come in contact with infected ones via feathers and feces. All these sources can act as potent sources of APEC infection [90,91]. APEC is a highly transmissible pathogen that can easily spread between chicks and affect the chick’s quality [92]. Christensen et al. [24] have previously discussed the effect of horizontal transmission in poultry. During hatch, handling, and transport of the chicks, the conditions are optimum for the spread and multiplication of APEC. To minimize this risk, it is recommended for the hatchery to be near the production facility. Also, an all-in all-out system would be an effective management strategy for preventing vertical and horizontal transmissions as there are chances of disinfection of the houses after each flock [24].

Another important aspect that facilitates the spread of *E. coli* at farms is vectors. Studies showed that APEC isolates were able to colonize vectors such as houseflies, wild birds, and even pigeons [92,93,94]. Of interest, beetles (*Alphitobius diaperinus*) were also found to be important in transmitting *E. coli.* Another parasite shown to be involved in the transmission is *Tetratrichomonas gallinarum* (Trichomonad spp.) [95,96]. Thus, focusing on controlling the vectors is essential to ensure biosecurity and prevent horizontal transmission [95]. Furthermore, the infected chicks during the first week of life may develop omphalitis, yolk sac infection, pericarditis, perihepatitis, and peritonitis. The body fluids of the infected chicks may be another important source of infection to other chicks in the flock. Moreover, one report suggests the transmission of APEC through feathers which can act as a source [97]. Controlling horizontal transmission is essential to control the increased first-week mortality and colibacillosis in broilers and breeders.

### 4.3. First-Week Mortality among Chicks

The first week of growth is crucial for a chick because it determines flock production ability, uniformity of the flock, and the age at processing [14]. APEC infections in chicks are characterized by acute and subacute septicemia. Death may occur due to acute septicemia resulting from infections in the yolk sac and respiratory system, while pericarditis, perihepatitis, and airsacculitis may develop as a result of sub-acute septicemia [14]. Chick quality can be assessed by looking into the first-week mortality pattern, which should be around 0 to 1% [98], and as per Aviagen standards, the first-week mortality should not exceed 0.7% [19]. Moreover, the European Union standards suggest that the first-week mortality is indicative of the welfare of the birds [99]. If mortality is more than 1%, it would affect the entire production system, necessitating its prevention. Even though several factors affect first-week mortality, such as breeder age, egg weight, genetics, hatchery conditions, feed quality, and house environment [100], the primary factor (50%) is infections due to various pathogens, and *E. coli* is a major pathogen in this group [101]. One study analyzed the role of *E. coli* in chick mortality 48–72 h after hatching and found around 70% of the dead chicks had colibacillosis symptoms [102]. Of interest, about 17% of first-week mortality among the chicks on average, *E. coli* was the major cause of death and was found to be the primary pathogen for the first-week mortality resulting from yolk sac infection [94,103,104]. As a secondary pathogen, *E. coli* along with *Enterococcus* infections was reported to be the cause of death in more than 50% of dead embryos in hatcheries. Furthermore, as a primary pathogen, a 19.46% prevalence of *E. coli* among dead embryos was observed [105]. 

The bacteria entering through the yolk sac infect the chicks that retain the yolk sac for a long period of time and cause the infection to peak 24–48 h after hatching, and at the end of about 2 weeks, the mortality may increase up to 10%–20%. Data show that, among the surviving population of chicks, about 5% may have stunted growth, while the development of the other birds will be normal [7,41]. Another important source of infection is the feed. Reports suggest that *E. coli* contaminates animal feed because of the poor-quality ingredients or the poor storage facilities [106]. Water is another important source of infection as *E. coli* is the indicator organism for fecal contamination [107]. Alarmingly, scientists have identified about 13% prevalence of *E. coli* in the water tanks of farms. It can act as a potential source of infection and can be avoided by strict biosecurity, proper disinfection, and management [106]. Moreover, floor eggs are another issue and it was reported to have high mortality among the chicks that came from floor eggs. For example, more than 60% *E. coli* prevalence was observed in chicks that came from floor eggs of a 60-week-old flock, but from normal eggs, it was only 40% [16]. Preventing mortality in the first week of life is important, and if the mortality rates are too high, the production facilities are recommended to reduce the number of birds in the subsequent cycle or until the cause of mortality is resolved. This is because the chicks are coming out of a conditioned environment in the hatchery to the brooder houses, where they may face many challenges, such as feed, thermoregulation, and fighting infections [47]. The first-week mortality is an important challenge to the broiler industry because it determines further growth and productivity. 

## 5. Prevention and Control of APEC in Breeders

### 5.1. Biosecurity and Management

Biosecurity is the primary factor pertaining to the prevention and control of APEC in broiler breeders. There should be more focus on the parent stock by periodic sampling and monitoring because from the breeders the isolates that cause the disease and anti-microbial resistance are transferred vertically. In addition, there are many primary bacterial and viral infections that compromise the immune barrier of birds and promote APEC infection as a secondary pathogen. These primary disease agents should be controlled by periodic monitoring and surveillance [31]. Furthermore, the spike males introduced to the flock must be from the same source and tested for important disease conditions, including APEC infections [33]. When the flock ages, the bird’s immune system gets compromised, eggshell thickness decreases, and the chick’s quality gets affected. Proper management of the flock, including culling and disinfecting the eggs, is critical in breeder flocks that are aged [15]. Additionally, there were reports of antibiotic-resistant (ESBL/AmpC) *E. coli* being recovered from the eggshells [84]. Formaldehyde fumigation is widely used for disinfection of hatching eggs but is carcinogenic and has a detrimental effect on the handler [108]. Alternative methods such as UV and spraying disinfectants could be practiced. For instance, a natural resin from honey bees called propolis has been found to be very effective without affecting hatchability or egg weight loss [109]. Moreover, hydrogen peroxide and essential oils like clove oil were also found to be effective without affecting hatchability or egg weight loss [110,111,112]. Disinfection procedures should be practiced regularly in the breeder house, the hatchery, and the brooder; practicing an all-in-all-out system might be helpful to some extent. Moreover, the floor eggs should be minimized either by discarding them or preventing their chance of contamination by proper cleaning or timely egg collection to reduce the contact time [16]. 

Care should be taken while transporting the chicks from one farm to another to prevent the transmission of bacteria by implementing good sanitation and biosecurity programs [16]. Importantly, controlling bacterial contamination in areas such as feed mills and packaging areas and prioritizing the biosecurity measures are critical in APEC control. Above that, culling weak chicks during the first week will reduce the loss due to treatment, and mortality is affected. It also helps to control the horizontal transmission [16]. Also, identifying the possible areas that are at risk of a breach in biosecurity is important and solutions should focus on controlling vertical transmission. 

### 5.2. Genetics

Genetics is another factor that could affect APEC infection in breeders. For instance, the fast-growing lineages are found to be more susceptible to APEC infections than the slow-growing lineages [113]. Furthermore, intense genetic selection for rapid growth and growth-related traits have increased the incidence of ascites in broilers and broiler breeders and birds with ascites are highly vulnerable to infectious agents [24]. As a result, Denmark is slowly trying to remove the fast-growing lineages from their production system and grow more slow-growing lines, and this could help to make the birds healthier [114]. Furthermore, scientists are developing breeds that are resistant to *E. coli* infections. They tried to identify the effect of maternal inheritance on disease resistance since the heritability of gamma globulins is 2.5 times higher in female parents than in male parents. Immunized parents are crossed in different ways and the effects are monitored. A significant maternal effect has been identified, but the reciprocal crosses had an intermediate effect for both parent lines. The reason for this needs to be investigated, and studies in this area would be impactful as there is evidence of genetic improvement that results in disease resistance [115].

### 5.3. Antibiotics

Various antibiotics are used to treat APEC infections in birds. Antibiotics such as tetracyclines, aminoglycosides, macrolides, sulphonamides, penicillins, cephalosporins, trimethoprim, quinolones, polymyxins, chloramphenicol, and lincosamides are routinely used for the treatment of colibacillosis [116]. However, recent studies have reported resistance of APEC towards most of these antibiotics. The improper and unrestricted use of antibiotics in animal agriculture might be a possible reason for this increase. For example, Colistin is an important drug in human therapeutics; however, Colistin-resistant *E. coli* species were recently found in poultry [13,117]. Previous studies which identified similarities between human and avian *E.coli* isolates necessitate strict antibiotic use regulations and monitoring [118,119]. Moreover, isolates collected from broiler breeders were resistant to many of the above-mentioned antibiotics (Table 2) [81,82]. Several countries, including the U.S., have restricted the use of antibiotics for non-therapeutic purposes (growth promoters) in animal production, and there is a high acceptance of the NAE production system in the market [21]. Constant monitoring of the antibiotic resistance pattern among disease-causing APEC isolates will be beneficial from a treatment standpoint.

### 5.4. Vaccination

Vaccination is one of the important strategies to limit APEC infections. Birds are vaccinated at day 1 and then around 12–14 weeks of age against *E. coli*. Developing efficient vaccines to prevent APEC infections is an important solution for limiting the use of antibiotics for treatment [120,121]. Moreover, preventing first-week mortality can be achieved through proper vaccination of the parent flocks [122,123]. Initially, only bacterins or inactivated vaccines were used against APEC, but after years of research, scientists developed live and subunit vaccines that became more popular [124]. Inactivated vaccines were used to study the possibility of vaccinating the broiler parents to provide protection and reduce the cost of vaccination for broilers [125]. The study also showed that antibodies derived from broiler parents could protect broiler chicks for up to 2 weeks before antibody protection was reduced [126]. Nobilis *E. coli* is a commercially available inactivated vaccine by Merck Sahrp & Dohme (MSD) Animal Health for active immunization of broiler breeders to provide passive immunization to broiler chickens. However, a study has shown a reduction in overall broiler breeder mortality but no effect on first-week chick mortality after Nobilis *E. coli* vaccination [127]. Also, there are limitations for inactivated vaccines as they are effective only against homologous challenges. Also, other factors such as frequency and route of administration and the age of the bird at vaccination also affect the efficacy [120]. Subunit vaccines that have been studied include iron-regulated outer membrane proteins-based, fimbriae-based, and increased serum survival proteins-based vaccines. Overall, the subunit vaccines are more effective than inactivated vaccines as they provide protection against heterologous challenges in addition to homologous challenges, but continuous follow-up studies in a commercial setting are necessary to confirm the field success of these vaccines [120]. Live attenuated vaccines are getting more acceptance as they are available for mass immunizations along with evidence of cell-mediated immunity rather than circulating antibodies alone [128]. Poulvac *E. coli* is a commercially available modified live vaccine by Zoetis. In addition to providing protection in layers and broilers, a trial conducted by a broiler breeder company in South Africa using this vaccine showed a decrease in mortality by curbing peritonitis [21,24]. Another trial was conducted in Denmark to evaluate the efficacy of the autogenous vaccine in the broiler breeder flock already vaccinated with Poulvac. However, there was no protection against homologous or heterologous infection even though there was an increased antibody response [129]. In ovo administration of Poulvac demonstrated that the air cell route is better than the amniotic route in terms of embryo mortality and hatchability. However, further studies are needed to confirm the long-term efficiency or protection of birds following in ovo administration of the vaccine [130].

Due to poor cross-protection of commercially available vaccines, efforts to develop autogenous vaccines against APEC in broiler breeders have been found to be more effective [131]. A study on autogenous vaccines reported that it influenced the selection of phylogroups after vaccination. They repressed most of the phylogroups but resulted in some strain shifts. For example, in one farm, the prevalence of phylogroup A decreased after vaccination even though it was present in the vaccine, whereas the prevalence of phylogroup F increased after vaccination which was not present in the vaccine [71]. Research on autogenous vaccines focusing on the target virulence factors which promote cross-protection would provide more efficient vaccines [72]. 

### 5.5. Probiotics and Prebiotics

Following restrictions on antibiotics, the use of alternatives to control APEC has been investigated for a long time. Probiotics are live non-pathogenic microbial feed supplements that provide health benefits such as protection from infectious agents, while prebiotics are non-digestible supplements that promote the growth of beneficial bacteria in the gut and thereby provide immunity [132,133]. Many scientists investigated the effect of *Lactobacillus plantarum* B1 probiotic in the feed of broilers and observed improvement in growth parameters along with a reduction in cecal *E. coli* output [132,133,134,135]. Additionally, the use of fructose oligosaccharide (FOS) along with *Lactobacillus plantarum* B1 was also tested, and improved performance was noted [134]. A challenge study using the O78 APEC strain to identify the effect of *Enterococcus faecalis-1* showed improvement in the performance and reduction in mortality among the challenged broiler birds [136]. Prebiotics such as FOS are also shown to be very effective in preventing bacterial infections, especially *E. coli.* Fructose oligosaccharide (FOS) can act as a substrate for *Bifidobacteria* which can synthesize bacteriocin that can suppress various pathogens [137]. Another important prebiotic is the mannose oligosaccharide (MOS) which can interfere with the bacterial adhesion mechanism. The most commonly used polysaccharide prebiotic in poultry is the partially hydrolyzed guar gum, which is basically an MOS [133]. Several probiotic and prebiotic mixes in the field are known to enhance immunity in the intestine as it is one of the most compromised areas that allow the spread of infection. Moreover, research should also focus on the reproductive system’s health because in broiler breeders, bacteria entering through the ascending route through the cloaca can infect the reproductive tract.

### 5.6. Bacteriophages

Bacteriophages are another possible source of treatment that could effectively prevent colibacillosis in chickens. They are viruses that target infectious bacteria without affecting the normal microflora [138,139,140,141]. A phage mixture using the bacteriophages SPR02 and DAF6 was administered through the air sac route against experimental infection of APEC O2 and showed a significant reduction in mortality in broilers [141]. Studies using phage cocktails supplemented through intra-tracheal and intra-venous routes also significantly reduced mortality and APEC load in the liver, lungs, and heart in broilers [138,139]. Phage-loaded chitosan particles administered orally have also shown a significant reduction in mortalities following experimental APEC infection in chickens. However, there are challenges in the practical application of these phage therapies because of the constraints in large-scale production and use in the poultry industry [141]. Studies on the effect of phage therapy in broiler breeders infected with APEC, including its effect on preventing vertical transmission, need to be carried out to develop phage as an alternative to antibiotic treatment in breeders.

### 5.7. Miscellaneous

Advanced techniques to control APEC infections are also promising. Innate immune stimulants are one among them which can stimulate the immune responses against pathogens. Cytosine–phosphodiester–guanine motifs are one of the important innate immune stimulants which activate pathogen-associated molecular patterns (PAMPs) [142,143]. Additionally, antimicrobial peptides are short positively charged peptides that act against most bacteria including those resistant to antibiotics. D-analog of chicken cathelicidin-2 is an important antimicrobial peptide tested against APEC in an in vivo challenge study in broilers [144,145]. Moreover, virulence inhibitors mainly cause the virulence mechanisms in bacteria unable to make virulence factors, thus, disarming the bacteria and not producing bacterial resistance [146,147]. In the future, in vivo and in vitro challenge studies in broiler breeders might reveal the efficiency of this alternative control method against broiler breeder APEC infections.

## 6. Conclusions

Broiler meat is one of the major sources of animal protein that is preferred globally, and broiler breeders play an inevitable role in sustainable broiler production; thus, the challenges in broiler breeder rearing should be minimized. APEC in broiler breeders is critical because it causes disease in breeders as well as is vertically transmitted from the hens’ reproductive tract through eggs to chicks and further horizontally transmitted between chicks. Moreover, evidence showing the transfer of antibiotic resistance genes vertically points out the importance of preventing and controlling APEC in broiler breeders. Using alternatives to antibiotics such as efficient vaccines, probiotics, prebiotics, and bacteriophages along with strict biosecurity and management practices could help limit the infection. Constant research focusing on the phenotypic and genotypic characterization of APEC isolates from broiler breeders is necessary because of the vast genetic diversity of this bacteria and its evolution as time passes.

## Figures and Tables

**Figure 1 pathogens-12-01280-f001:**
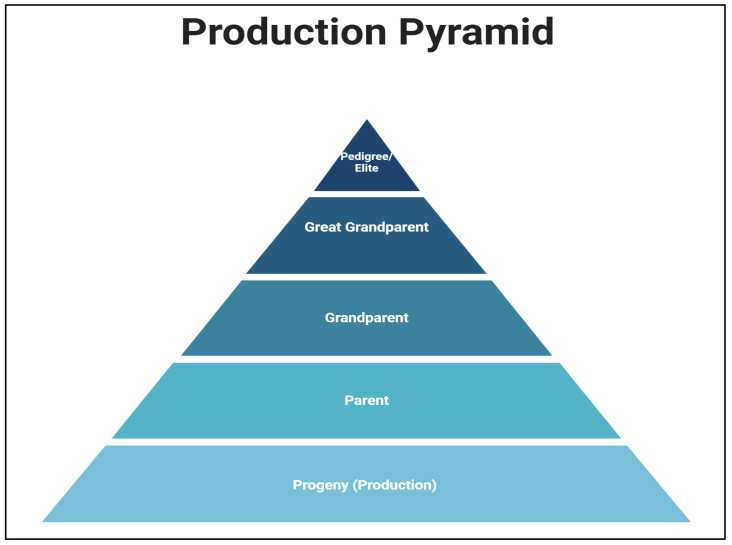
Image showing the production pyramid in the poultry industry. The highest level of the pyramid is the elite or the pedigree group and the lowest level is the progeny or the production farm.

**Figure 2 pathogens-12-01280-f002:**
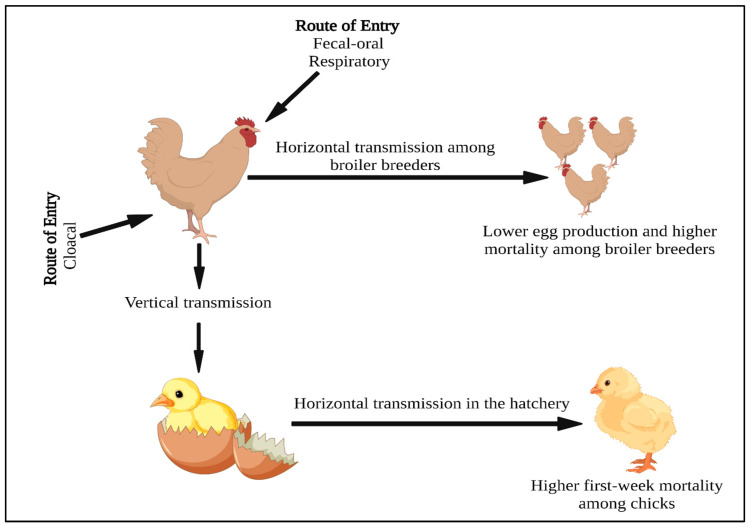
Image showing the APEC route of entry, vertical transmission to the chicks, and horizontal transmission in the hatchery and among broiler breeders.

**Table 1 pathogens-12-01280-t001:** Genotypic virulence characteristics of *E. coli* strains isolated from broiler breeders from different geographical locations.

Location	*E. coli* Isolated from	Virulence-Associated Genes Identified	Serogroups Identified	Phylogroups Identified	Source
United States	Broiler breeder	*iroN*, *iutA*, *iss*, *hlyF*, *ompT*, *papC*, *tsh*, *ibeA*, *astA*, and *cva/cvi*	O88, O8, O25, O115, O166, O161, O1, O55, H9, H21, H4, H7	B1, B2, D	[20]
Colombia	Broiler breeder	N/A	O1, O2, O5, O36, O45, O53 O78	N/A	[51]
Thailand	Broiler breeder	*iroN*, *iutA*, *iss*, *hlyF*, and *ompT*	N/A	N/A	[60]
Australia	Broiler breeder	*iroN*, *iutA*, *iss*, *hlyF*, and *ompT*	N/A	N/A	[60]
Korea	Broiler breeder	*iroN*, *iutA*, *iss*, *hlyF*, and *ompT*	N/A	N/A	[61]
Canada	Broiler breeder *	*sitA*, *iroN*, and *iutA*, *iss*, *ompT*, *etsB*, *cvaC*, *ireA*	N/A	N/A	[62]
Mexico	Broiler breeder, Hatchery	N/A	O19, O18, O8, O78, O6, O9, O52, O125	N/A	[68]
Croatia	Broiler breeder	N/A	N/A	A, B1, B2, C, D, E, F	[71]
Korea	Broiler breeder *	*fimC*, *tsh*, *fyuA*, *irp2*, *iucD*, *cav/cvi*, *iss*, *astA*, *vat*	O1, O6, O8, O15, O18, O25, O26, O28, 078, O111, O112, O115, O125, O119, O126, O167	N/A	[74]
Denmark	Broiler breeder	*papAH*, *papEF*, *papGII*, *ireA*, *iroN*, *iucD*, *yuA*, *kpsMTK1*, *kpsMTII*, *malX[PAI]*, *fimH*, *iss*, *traT*, *sitA*, *vat*, *astA*, *cvi/cva*, *ibeA*, *tsh*, *ompT*	N/A	A, B1, B2, D	[47]

N/A, data not available. * The samples used for these studies are pooled from broiler breeders, broilers, and layers.

**Table 2 pathogens-12-01280-t002:** Antibiotic resistance patterns observed among *E. coli* isolates from broiler breeder/broiler chicks reported worldwide.

Location	*E. coli* Isolated from	Antibiotic Resistance Genes Identified	Resistant Antibiotics Identified by Susceptibility Testing	Multi-Drug Resistance Percentage/%	Source
Algeria	Broiler breeders	*blaCTX-M-1*, *blaSHV-12*, *blaTEM-1*	NAL, UB, CIP, LVX, AMP, AMC, XNL, TET, TPM, SXT, NEO, CHL	89.18	[17]
United States	Broiler breeders	*blaTEM*, *blaCTX-M-1*, *aph3IA*, *aadA*, *tetA*, *sul1*	TET, STM, KAN, GEN, SXT	10.7	[20]
Italy	Broiler breeder	-	AMX, ENR, TET, SXT	100	[45]
Belgium	Broiler breeder (roosters)	-	CTX, XNL, AMX, CFM, TMP, NAL, CZM	N/A	[56]
Thailand	Broiler breeder	-	AMX, CEF, CHL, CIP, ENR, FLO, GEN, NAL, SXT, TET	60.71	[60]
Australia	Broiler breeder	-	AMX, CEF, CHL, CIP, ENR, FLO, GEN, NAL, SXT, TET	10.71	[60]
Korea	Broiler breeder	*blaTEM-1*, *blaTEM-135*, *blaTEM-176*, *blaCTX-M-1*, *blaCTX-M-55*, *qnrS*, *qnrB*, *qnrA*, *cmlA*, *catA1*, *aac(6’)-Ib*, *aac(3)-II*, *tetA*, *tetB*, *tetC*	TET, NAL, AMP, CEP, SXT, CHL, CZN, GEN, LVX	81.6	[61]
Egypt	Broiler chicks—first week	-	AMP, AUG, CTX, CST, TET, DOX, GEN, NEO, OFX, ENR, CIP, FLO	N/A	[75]
Algeria	Broiler breeder	*blaCTX-M-1*, *qnrS1* *	AMP, TET, PRL, CZN, CXM, NAL	89	[79]
Denmark	Broiler breeder/Broiler	-	AMP, NAL, CIP	N/A	[80]
Algeria	Broiler breeder	-	TET, NAL, AUG, AMP, SXT, ENR, NIT, NEO, CHL, GEN	100	[81]
Switzerland	Broiler breeder	*GyrA*, *ParC* *	ENR	N/A	[82]
Sweden	Broiler breeder	*blaCMY-2*	-	N/A	[83]
Indonesia	Broiler breeder/Layer breeder	*tetA*, *tetB*	AMP, NAL, TET, OTC, GEN, CHL	69	[84]
France	Broiler chicks—first week	-	TET, AMX, SXT, NAL, GEN	N/A	[85]

TET, tetracycline; NAL, nalidixic acid; AUG, amoxicillin–clavulanic acid; AMP, ampicillin; SXT, sulfamethoxazole and trimethoprim; ENR, enrofloxacin; NIT, nitrofurantoin; NEO, neomycin; CHL, chloramphenicol; GEN, gentamycin; CTX, cefotaxime; CST, colistin; DOX, doxycycline; OFX, ofloxacin; CIP, ciprofloxacin; FLO, florfenicol; CEF, cefalexin; PRL, piperacillin; CZN, cefazolin; CXM, cefuroxime; CEP, cephalothin; OTC, oxytetracycline; UB, flumequine; LVX, levofloxacin; AMC, amoxicillin; XNL, ceftiofur; TPM, trimethoprim; CZM, ceftazidime; STM, streptomycin; KAN, kanamycin. N/A, data not available. * Vertically transmitted.

**Table 3 pathogens-12-01280-t003:** Comprehensive data from different geographical locations showing vertical transmission of *E. coli* from broiler breeders to chicks.

Location	Source	Factors/Techniques Used to Identify Vertical Transmission	Source
Denmark	Broiler breeders, Eggs	Bacterial count and lesions	[15]
Denmark	Broiler breeders, Live chicks	PGFE and MLST *	[16]
Finland	Broiler breeders, Eggs, Poultry house	Antibiotic resistance genes (*ESBL/pAmpC*)	[18]
Italy	Broiler breeders, Dead chicks	Serogrouping (O78 and O139)	[44]
Denmark	Broiler breeders, Broilers	Antibiotic resistance (AMP and NAL) *	[80]
Sweden	Broiler breeders, Broilers, Hatchery	Antibiotic resistance gene (*pAmpC*)	[83]
Switzerland	Broiler breeders, Live chicks, Environmental sample	Antibiotic resistance gene (*blaCTX-M-1)*	[85]
United States	Broiler breeders	Ec (NAL) *	[88]
Netherlands	Broiler breeders, Environmental sample	Antibiotic resistance genes (*ESBL/AmpC*)	[89]

* PGFE, pulsed-field gel electrophoresis; MLST, multilocus sequence typing; AMP, ampicillin; NAL, nalidixic acid; Ec (NAL), *Escherichia coli* in nalidixic acid medium.

## Data Availability

Not applicable.

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
