# Peer review of "Avian Pathogenic Escherichia coli (APEC) in Broiler Breeders: An Overview"

_pathogens, 2023, doi:10.3390/pathogens12111280_

Round 1

Reviewer 1 Report

Comments and Suggestions for Authors

General Comments:

There are multiples issues in this article that strictly need to be addressed with other revisions. The article seems to be repetition of these two articles

 1) https://doi.org/10.3390/pathogens10040467

2) https://doi.org/10.1080/03079457.2020.1845300

1) First of all, it should be cleared in the review that, is this only a literature review or a scientific data based? Because multiple parameters are missing while quoting about the previous studies, including percent positivity of the genes among APEC isolates. And it seems like a general discussion of the APEC.

2) While talking about the virulence associated genes (VAGs) and Antimicrobial resistance (AMR) genes, it must be clear about the method of genetic determination and should mention it clearly. Either the previous authors used conventional methods or modern methods to detect the genes among APEC isolates? Was it phenotypic detection? Or genotypic by polymerase chain reaction (PCR) or whole genome sequence (WGS) data analysis? And in future what kind of inventions we need regarding upgrading the technology to detect APEC isolates?

3) Define a pathotype. Is there any classical lesion of colibacillosis? Previous studies have differentiated between Avian Fecal E. coli (AFEC) and Avian Pathogenic E. coli (APEC) on some basis. This article ld describe these properties. Also not all APEC are pathogenic, there is some prediction about highly pathogenic strains of APEC. What is that?

4) The data on sequence types (STs) of APEC among broilers is also missing.

First Paragraph of “Introduction” was a good start about US poultry production and values. But it ends with the description of challenges that contain references of studies from other countries, Canada, Japan and Bangladesh (Ref. 5, 6)

Line No. 30-31: Author is suggested to add challenges faced by USA poultry industry and also economic losses annually.

Line No. 31-37: “Broiler breeder are . . . . . . “ Author is suggested to make it a separate paragraph of general challenges faced by the broiler poultry industry worldwide.

Line No. 39: Add word “strains” in the sentence… “However, some of the strains can act as…….”

Line No. 40: A little classification of Extra-intestinal pathogenic E. coli (ExPEC), showing APEC as pathotype, is suggested here.

Line 43-45: Author is suggested to write different major types of pathological manifestations of colibacillosis separately (localized and/or systemic). Check Table No. 18.6 of Nolan Chapter, https://doi.org/10.1002/9781119421481.ch18

Line No. 49-50: Sentence needs some scientific attention. Correct or Re-phrase it.

Line No. 57: Add word “contaminated egg shells can act as potential source of ….)

Line No. 65: Add word “antibiotic resistant bacterial species……..

Line No. 66: The author is suggested to link and cite following studies here:
https://doi.org/10.1016/j.aninu.2018.03.002
https://doi: 
10.3389/fvets.2023.1139908

Line No. 75: Addition of a Figure showing a typical organization of broiler breeding in the form of pyramid structure is needed here. Because author is mentioning the “production pyramid” multiple times throughout in the article.

The Second heading of the article “2. Broiler breeder industry and its’ challenges” seems to be a RE-START regarding “Broiler Breeders” mentioning about the body weight, breeding companies, the production pyramid etc . Which confuses a reader because starting from US broiler production we were moving towards the challenges of poultry industry due to APEC AMR, Control etc., step by step (Line No. 59-75). The author is suggested to keenly observe the sequence of literature written under heading No. 1 & 2 of the article and merge it sequence wise. First write all about USA, then the breeding companies, then about broiler breeder and the production pyramid, general challenges of broiler poultry industry . . all in 1 heading “Introduction”

Line No. 151-155: The information quoted here seems to be repeated. Correct/Re-phrase it.

Line No: 199: Add abbreviation of virulence associated genes (VAGs)

Line No. 200-220: Please re-check the statements. Clear that you are adding data only about “broiler breeders” or “broilers” or both. There must be uniformity in all article. Similar data about APEC from broilers of different geographical locations is missing. The author is suggested to add the following studies and also cite in the Table 1. Also ADD MORE studies, there must be some from UK, Austrailia, Iran etc. OR Limit your review to some geographical area.

Brazil: Barbieri et al., 2015 (https://doi.org/10.1089/fpd.2014.1815)
Nepal: Subedi et al., 2018 (https://doi.org/10.1186/s12917-018-1442-z)
Pakistan: Azam et al., 2019 (https://doi.org/10.1007/s11250-019-01823-3)
Jordan: Ibrahim et al., 2019 (https://doi.org/10.1186/s12917-019-1901-1)
Pakistan: Azam et al., 2020 (https://doi.org/10.1016/j.vetmic.2020.108766)
USA: Newman et al., 2021 (https://doi.org/10.7717/peerj.11025)
Italy: Apostolakos et al., 2021 (https://doi.org/10.3389/fvets.2021.737720)
Poland: Wilczynski et al., 2022 (https://doi.org/10.3390/ani12091090
Thailand: Thomrongsuwannakij et al., 2022: (https://doi.org/10.1016/j.psj.2021.101527)
Brazil: Barbosa et al., 2023 (https://doi.org/10.1007/s42770-023-01023-0)

Table: 2 Column “Multiple drug resistance identified” seems to be an EXTRA. As most of the studies has identified it so it is obvious from resistance patterns of isolates. This column should be deleted. Also add MDR data from the above mentioned studies from different geographical areas of the World.

Line No. 295: A new heading is suggested to be added here. As now author is talking about resistance towards heavy metals.

Line No. 309-310: There is a need to add some study about environmental modifiers that can reduce the Ammonia Levels at poultry farms and aid in improvement in the production parameters of broilers.

Line No. 313-451: Heading No. 4. The literature under subheadings No. 4.1 to 4.3 needs to be comprehensive, to the point and informative. Remove repeated and unnecessary information from here.

Table:3 Formatting seems much line spacing.

Line No. 452-621: Heading No. 5. The literature under subheadings No. 5.1 to 5.7 are also extensive. Needs to be comprehensive. 

Line No. 511: Skip mentioning colistin resistant E. coli from swine. The article is about APEC from broilers.

Line 511, Please cite study of Zulqarnain et al., 2021 http://www.pvj.com.pk/pdf-files/41_2/284-288.pdf

Line No. 534: What is MSD? Write its full name, then abbreviation in bracket.

Line No. 589: The author is suggested to add the study http://www.pvj.com.pk/pdf-files/41_2/269-273.pdf

Line No. 613: Some other relevant citations can be taken from following review https://www.ncbi.nlm.nih.gov/pmc/articles/PMC8069529/

Comments on the Quality of English Language

Minor editing of English language required

Author Response

 Point 1: There are multiples issues in this article that strictly need to be addressed with other revisions. The article seems to be repetition of these two articles.

 1) https://doi.org/10.3390/pathogens10040467

2) https://doi.org/10.1080/03079457.2020.1845300

Response 1: Thank you for the feedback. However, the focus of this review is APEC specifically in broiler breeders, and the above-mentioned articles concentrate more on APEC infections in poultry in general or with an emphasis on broilers. It is important to understand APEC isolates from broiler breeders because of the involvement of vertical transmission and that may result increased first week morthality among the chicks which is a concern to the industry. Also, there are only limited studies on this field and thus, it necessitates the importance of a review article which could provide a comprehensive idea about the characteristics of APEC isolates from broiler breeeders. Therefore,  in this review, we gathered information that can apply to broiler breeders from the available resources. This review will be extremely beneficial to poultry scientists and veterinarians focusing on APEC mitigation in broiler breeders, chicks, and broilers.

Point 2:  First of all, it should be cleared in the review that, is this only a literature review or a scientific data based? Because multiple parameters are missing while quoting about the previous studies, including percent positivity of the genes among APEC isolates. And it seems like a general discussion of the APEC.

Response 2:  Thanks for the comments. This is a scientific data based review article and throughout the manuscript we have included the percentage positivity of the genes from the studies that are available. We have added more specific details according to the reviewer's comments. Please refer to lines 226-234, line 236, line 250, line 311, line 342, line 359, lines 402-403, lines 447-475 in the revised manuscript with track changes.

Point 3:  While talking about the virulence associated genes (VAGs) and Antimicrobial resistance (AMR) genes, it must be clear about the method of genetic determination and should mention it clearly. Either the previous authors used conventional methods or modern methods to detect the genes among APEC isolates? Was it phenotypic detection? Or genotypic by polymerase chain reaction (PCR) or whole genome sequence (WGS) data analysis? And in future what kind of inventions we need regarding upgrading the technology to detect APEC isolates?

Response 3: As per the reviewer’s comment, the methods of genetic determination have been added. Please refer to lines 224-245, line 313 in the revised manuscript with track changes. In the future, we may need to have more specific genes/ virulence factors that are conservative to identify the organisms using multiplex PCRs or other sequencing methods. The advancement in whole genome sequencing and increased throughput might possibly make this happen. The information is added in line 242-246.  

Point 4: Define a pathotype. Is there any classical lesion of colibacillosis? Previous studies have differentiated between Avian Fecal E. coli (AFEC) and Avian Pathogenic E. coli (APEC) on some basis. This article ld describe these properties. Also not all APEC are pathogenic, there is some prediction about highly pathogenic strains of APEC. What is that?

Response 4: The details about the APEC pathotype (including the lesions) have been included in the manuscript according to the reviewer’s comment. Please refer to lines 48-58 in the revised manuscript with track changes.

Point 5: The data on sequence types (STs) of APEC among broilers is also missing.

Response 5: Thanks for pointing this out. We have added the available information about the sequence types among broiler breeder APEC isolates in the manuscript. Please refer to lines 256 to 261 in the revised manuscript with track changes.   

Point 6: First Paragraph of “Introduction” was a good start about US poultry production and values. But it ends with the description of challenges that contain references of studies from other countries, Canada, Japan and Bangladesh (Ref. 5, 6)

Response 6: Thanks for pointing it out. The issue with References has been addressed. Please refer to line 43 in the revised manuscript with track changes. 

Point 7: Line No. 30-31: Author is suggested to add challenges faced by USA poultry industry and also economic losses annually.

Response 7: As per the reviewer comment,  challenges faced by US poultry industry is added to the manuscript’s Introduction section. Please refer to lines 30 to 38 in the revised manuscript with track changes. 

Point 8: Line No. 31-37: “Broiler breeder are . . . . . . “ Author is suggested to make it a separate paragraph of general challenges faced by the broiler poultry industry worldwide.

Response 8: As per the reviewer comment, necessary changes has been made. Please refer to lines 41 to 47 in the revised manuscript with track changes. 

Point 9: Line No. 39: Add word “strains” in the sentence… “However, some of the strains can act as…….”

Response 9: As per the reviewer’s comment, necessary change has been made. Please refer to line 49 in the revised manuscript with track changes.

Point 10: Line No. 40: A little classification of Extra-intestinal pathogenic E. coli (ExPEC), showing APEC as pathotype, is suggested here.

Response 10: As per the reviewer’s comment, the classification of ExPEC has been added. Please refer to lines 52-53 in the revised manuscript with track changes.

Point 11: Line 43-45: Author is suggested to write different major types of pathological manifestations of colibacillosis separately (localized and/or systemic). Check Table No. 18.6 of Nolan Chapter, https://doi.org/10.1002/9781119421481.ch18

Response 11: As per the reviewer’s comment, pathological manifestations have been separated into local and systemic. Please refer to lines 55-58 in the revised manuscript with track changes.

Point 12: Line No. 49-50: Sentence needs some scientific attention. Correct or Re-phrase it..

Response 12: As per the reviewer’s comment, necessary changes have been made. Please refer to line 62-63 in the revised manuscript with track changes.

Point 13: Line No. 57: Add word “contaminated eggshells can act as potential source of ….)

Response 13: As per the reviewer’s comment, necessary change has been made. Please refer to line 70 in the revised manuscript with track changes.

Point 14: Line No. 65: Add word “antibiotic resistant bacterial species……..

Response 14: As per the reviewer’s comment, necessary change has been made. Please refer to lines 79-80 in the revised manuscript with track changes.

Point 15: Line No. 66: The author is suggested to link and cite following studies here:
https://doi.org/10.1016/j.aninu.2018.03.002
https://doi: 10.3389/fvets.2023.1139908

Response 15: Thanks for the suggestion. We have added the references and cited them in the manuscript. Please refer to line 81 in the revised manuscript with track changes.

Point 16: Line No. 75: Addition of a Figure showing a typical organization of broiler breeding in the form of pyramid structure is needed here. Because author is mentioning the “production pyramid” multiple times throughout in the article.

Response 16: Thanks for the suggestion. As per the reviewer's comment, a new figure (Fig. 1) has been added to the manuscript and is cited in the text in line 117 of the revised manuscript with changes.

Point 17: The Second heading of the article “2. Broiler breeder industry and its’ challenges” seems to be a RE-START regarding “Broiler Breeders” mentioning about the body weight, breeding companies, the production pyramid etc . Which confuses a reader because starting from US broiler production we were moving towards the challenges of poultry industry due to APEC AMR, Control etc., step by step (Line No. 59-75). The author is suggested to keenly observe the sequence of literature written under heading No. 1 & 2 of the article and merge it sequence wise. First write all about USA, then the breeding companies, then about broiler breeder and the production pyramid, general challenges of broiler poultry industry . . all in 1 heading “Introduction”.

Response 17: Thank you for the suggestions. The first heading “Introduction” is an opening section providing a general overview of the different headings/topics covered in this review article. Specific details are mentioned under each heading. Under the second heading “broiler breeder industry and its challenges” we discussed in detail the U.S poultry industry, broiler breeder industry and various challenges faced by the industry, and in the following headings everything including APEC in broiler breeders, AMR, virulence genes, and control are discussed in detail.

Point 18: Line No. 151-155: The information quoted here seems to be repeated. Correct/Re-phrase it.

Response 18: As per the reviewer’s comment, necessary changes have been made. Please refer to lines 175-176  in the revised manuscript with track changes.

Point 19: Line No: 199: Add abbreviation of virulence associated genes (VAGs)

Response 19: As per the reviewer’s comment, abbreviation has been added to the manuiscript. Please refer to line 222 in the revised manuscript with track changes.

Point 20: Line No. 200-220: Please re-check the statements. Clear that you are adding data only about “broiler breeders” or “broilers” or both. There must be uniformity in all article. Similar data about APEC from broilers of different geographical locations is missing. The author is suggested to add the following studies and also cite in the Table 1. Also ADD MORE studies, there must be some from UK, Austrailia, Iran etc. OR Limit your review to some geographical area.

Brazil: Barbieri et al., 2015 (https://doi.org/10.1089/fpd.2014.1815)
Nepal: Subedi et al., 2018 (https://doi.org/10.1186/s12917-018-1442-z)
Pakistan: Azam et al., 2019 (https://doi.org/10.1007/s11250-019-01823-3)
Jordan: Ibrahim et al., 2019 (https://doi.org/10.1186/s12917-019-1901-1)
Pakistan: Azam et al., 2020 (https://doi.org/10.1016/j.vetmic.2020.108766)
USA: Newman et al., 2021 (https://doi.org/10.7717/peerj.11025)
Italy: Apostolakos et al., 2021 (https://doi.org/10.3389/fvets.2021.737720)
Poland: Wilczynski et al., 2022 (https://doi.org/10.3390/ani12091090
Thailand: Thomrongsuwannakij et al., 2022: (https://doi.org/10.1016/j.psj.2021.101527)
Brazil: Barbosa et al., 2023 (https://doi.org/10.1007/s42770-023-01023-0)

Response 20: Thank you for pointing it out. The article focusses ONLY on “broiler breeders” and hence, to avoid any confusion to the readers, the information about broilers have been removed from Table 1.

Point 21: Table: 2 Column “Multiple drug resistance identified” seems to be an EXTRA. As most of the studies has identified it so it is obvious from resistance patterns of isolates. This column should be deleted. Also add MDR data from the above mentioned studies from different geographical areas of the World.

Response 21: As per the reviewer’s comment, necessary changes have been made. Please refer to Table.2 in the revised manuscript with track changes.

Point 22: Line No. 295: A new heading is suggested to be added here. As now author is talking about resistance towards heavy metals.

Response 22: As per the reviewer’s comment, a new heading has been added for the metal resistance part. Please refer to line 332 in the revised manuscript with track changes.

Point 23: Line No. 309-310: There is a need to add some study about environmental modifiers that can reduce the Ammonia Levels at poultry farms and aid in improvement in the production parameters of broilers.

Response 23: Thanks for the suggestion. However, as mentioned above, the focus of this article is APEC infections in broiler breeders and adding more details pertaining to broilers might confuse the readers and hence, it is not added in the manuscript.

Point 24: Line No. 313-451: Heading No. 4. The literature under subheadings No. 4.1 to 4.3 needs to be comprehensive, to the point and informative. Remove repeated and unnecessary information from here.

Response 24: As per the reviewer's comment, necessary changes have been made in subsections 4.1 to 4.3.

Point 25: Table formatting seems much line spacing

Response 25: Thanks for the feedback. The line spacings for Table 3 has been adjusted.

Point 26: Line No. 452-621: Heading No. 5. The literature under subheadings No. 5.1 to 5.7 are also extensive. Needs to be comprehensive. 

Response 26: As per the reviewer's comment, necessary changes have been made in subsections 5.1 to 5.7.

Point 27: Line No. 511: Skip mentioning colistin resistant E. coli from swine. The article is about APEC from broilers

Response 27: As per the reviewer's comment, the swine part of the colistin resistance has been removed from the manuscript. Please refer to line 553 in the revised manuscript with track changes.

Point 28: Line 511, Please cite study of Zulqarnain et al., 2021 http://www.pvj.com.pk/pdf-files/41_2/284-288.pdf

Response 28: As per the reviewer's comment, the reference has been added to the manuscript. . Please refer to line 553 in the revised manuscript with track changes.

Point 29: Line No. 534: What is MSD? Write its full name, then abbreviation in bracket.

Response 29: As per the reviewer's comment, necessary change has been made. Please refer to line 579 in the revised manuscript with track changes.

Point 30: Line No. 589: The author is suggested to add the study http://www.pvj.com.pk/pdf-files/41_2/269-273.pdf

Response 30: Thank you for the suggestion However, as mentioned above, the focus of this article is APEC infections in broiler breeders and adding more details pertaining to broilers might confuse the readers and hence, this study is not cited in the manuscript.

Point 31: Line No. 613: Some other relevant citations can be taken from the following review https://www.ncbi.nlm.nih.gov/pmc/articles/PMC8069529/

Response 31: : As per the reviewer's comment, relevant references have been added to the manuscript. Please refer to lines 659-665 in the revised manuscript with track changes.

Reviewer 2 Report

Comments and Suggestions for Authors

This is a comprehensive review with a lot of good information.  Biggest problem is the unscientific language choices.  Most of these can be easily fixed, although there are a few areas where additional explanation/clarification is required.

L12 would suggest 'APEC cuases'

L36 timely solutions

L39 some E. coli

L41 delete 'are the ones that'

L72 delete 'from time to time'

L76 understated is not the correct term please revise.

L93 either major or primary, not both

L102 'such as' works better than 'like'

L111 too many brackets

L124 not a sentence, please revise.

L133 delete 'from time to time'

L345 help to limit disease in broiler breeders

L134 Vaccines are further discussed...

L170 delete 'in breeders' have enough breeders already in this sentence

L175 orchitis may occur

L179-181 delete 'rooster' as not needed.  Only roosters would be producing semen

L190 What is a spike male?  This needs to be better explained.

L209 another pattern of prevalent genes

L211 most prevalent

L218-219 This does not make sense. Don't the 5 genes identified by Johnson et al. also predict APEC in broiler breeders?  Please clarify.

L221-229 This section is confusing. Phylogroups are varying how?  Is B2 the most highly pathogenic phylogroup?  Table 1 does nothing to clarify phylogroups.  Do certain serogroups or virulence genes indicate phylogroups?  In table 1 only first study has all information so not easy to understand.

L245 per isolate

L270 delete 'highly'

L282 Currently, the broiler industry...

L303 delete 'before'

L318 genuine vertical transfer??? Is there fake vertical transfer?  Please clarify.

L332 delete 'In a study'

L341 The concern should be monitored??? Please re-phrase.

L376, 382 and elsewhere.  What is fluff?  The feathers on chicks?  Please use a more scientific term.

L430 'increase' is better than 'rise up'

L447 until not 'till'.  Till = plow the field.

L456 typo

L458 that compromise the

L484 is affected

L495 are developing breeds

L501 delete 'highly

L528 this sentence does not make sense

L533 up to 2 weeks before antibody protection was reduced.

L567 it = the vaccine?

L574 probiotics are not necessarily inactivated.  Generally, they are  live non-pathogenic bacteria.

L584 are also shown

L597 are SPR02 and DAF6 the phages?  Unclear.

L615 including those resistant to antibiotics

L622 Should the US also be following Denmark and selecting for disease resistance instead of single trait selection on growth rate?  Important to discuss.

Comments on the Quality of English Language

Numerous word choice issues - outlined above.

Author Response

This is a comprehensive review with a lot of good information.  The biggest problem is the unscientific language choices.  Most of these can be easily fixed, although there are a few areas where additional explanation/clarification is required.

 Point 1: L12 would suggest 'APEC causes.’

Response 1: Thanks for the suggestion. As per the comment, the necessary change has been made. Please refer to line 12 in the revised manuscript with track changes.

Point 2: L36 timely solutions.

Response 2: As per the comment, necessary change has been made. Please refer to line 46 in the revised manuscript with track changes.

Point 3: L39 some E. coli

Response 3: As per the comment, necessary change has been made. Please refer to line 49 in the revised manuscript with track changes.

Point 4: L41 delete 'are the ones that.’

Response 4: As per the comment, necessary change has been made. Please refer to line 51 in the revised manuscript with track changes.

Point 5: L72 delete 'from time to time.’

Response 5: As per the comment, necessary change has been made. Please refer to line 86 in the revised manuscript with track changes.

Point 6: L76 understated is not the correct term please revise.

Response 6: As per the comment, necessary change has been made. Please refer to line 92 in the revised manuscript with track changes.

Point 7: L93 either major or primary, not both.

Response 7: As per the comment, necessary change has been made. Please refer to line 111 in the revised manuscript with track changes.

Point 8: L102 'such as' works better than 'like.’

Response 8: As per the comment, necessary change has been made. Please refer to line 120 in the revised manuscript with track changes.

Point 9: L111 too many brackets.

Response 9: As per the comment, removed the extra bracket. Please refer to line 129 in the revised manuscript with track changes.

Point 10: L124 not a sentence, please revise.

Response 10: As per the comment, the sentence has been modified. Please refer to lines 142 to 145 in the revised manuscript with track changes.

Point 11: L133 delete 'from time to time'.

Response 11: As per the comment, necessary change has been made. Please refer to line 154 in the revised manuscript with track changes.

Point 12: L134 help to limit disease in broiler breeders.

Response 12: As per the comment, necessary change has been made. Please refer to line 155-156 in the revised manuscript with track changes.

Point 13: L134 Vaccines are further discussed...

Response 13: As per the comment, necessary change has been made. Please refer to line 156 in the revised manuscript with track changes.

Point 14: L170 delete 'in breeders' have enough breeders already in this sentence.

Response 14: As per the comment, necessary change has been made. Please refer to line 193 in the revised manuscript with track changes.

Point 15: L175 orchitis may occur.

Response 15: As per the comment, necessary change has been made. Please refer to line 198 in the revised manuscript with track changes.

Point 16: L179-181 delete 'rooster' as not needed.  Only roosters would be producing semen.

Response 16: As per the comment, necessary change has been made. Please refer to lines 201 to 204 in the revised manuscript with track changes.

Point 17: L190 What is a spike male?  This needs to be better explained.

Response 17: As per the comment, spike male has been explained. Please refer to line 213-214 in the revised manuscript with track changes.

Point 18 L209 another pattern of prevalent genes.

Response 18: As per the comment, necessary change has been made. Please refer to line 233-234 in the revised manuscript with track changes.

Point 19: L211 most prevalent.

Response 19: As per the comment, necessary change has been made. Please refer to line 235 in the revised manuscript with track changes.

Point 20: L218-219 This does not make sense. Don't the 5 genes identified by Johnson et al. also predict APEC in broiler breeders?  Please clarify.

Response 20: The five genes (iroN, iss, iutA, hlyF, and ompT) identified by Johnson et al primarily predict APEC in broilers. However, these five genes were not always identified among APEC isolated from broiler breeders. Therefore, it is important to identify the specific APEC markers in broiler breeders for accurate disease diagnosis. This clarification is included in lines 240 to 247 in the revised manuscript with track changes.

Point 21: L221-229 This section is confusing. Phylogroups are varying how?  Is B2 the most highly pathogenic phylogroup?  Table 1 does nothing to clarify phylogroups.  Do certain serogroups or virulence genes indicate phylogroups?  In table 1 only first study has all information so not easy to understand.

Response 21: Thank you so much for the feedback. According to the new Clermont classification there are 8 phylogroups, A, B1, B2, C, D, E, F and clade 1 while old classification had only 4 phylogroups, A, B1, B2, and D. Majority of the APEC isolates fall under B2 phylogroup irrespective of the type of classification used, however, rest of the phylogroups shifted from A to C and D to E or F when new Clermont classification was used instead of old classification. As there are only limited studies using broiler breeder isolates, we suggest considering the new Clermont classification and changing patterns if any, to get a comprehensive idea about broiler breeder APEC isolates. Please refer to lines 249-257 in the revised manuscript with track changes.

Point 22: L245 per isolate

Response 22: As per the comment, necessary change has been made. Please refer to line 281 in the revised manuscript with track changes.

Point 23: L270 delete 'highly'

Response 23: As per the comment, necessary change has been made. Please refer to line 306 in the revised manuscript with track changes.

Point 24: L282 Currently, the broiler industry...

Response 24: As per the comment, necessary change has been made. Please refer to line 319 in the revised manuscript with track changes.

Point 25: L303 delete 'before'

Response 25: As per the comment, necessary change has been made. Please refer to line 341 in the revised manuscript with track changes.

Point 26: L318 genuine vertical transfer??? Is there fake vertical transfer?  Please clarify.

Response 26: Thanks for pointing it out. As per the comment, the word genuine has been removed. Please refer to line 357 in the revised manuscript with track changes.

Point 27: L332 delete 'In a study'

Response 27: As per the comment, necessary change has been made. Please refer to line 371 in the revised manuscript with track changes.

Point 28: L341 The concern should be monitored??? Please re-phrase.

Response 28: As per the comment, necessary change has been made. Please refer to line 380 in the revised manuscript with track changes.

Point 29: L376, 382 and elsewhere.  What is fluff?  The feathers on chicks?  Please use a more scientific term.

Response 29: As per the comment, necessary changes have been made. Please refer to lines 415 to 439 in the revised manuscript with track changes.

Point 30: L430 'increase' is better than 'rise up'

Response 30: As per the comment, necessary change has been made. Please refer to line 469 in the revised manuscript with track changes.

Point 31: L447 until not 'till'.  Till = plow the field.

Response 31: As per the comment, necessary change has been made. Please refer to line 486 in the revised manuscript with track changes.

Point 32: L458 that compromise the

Response 32: As per the comment, necessary change has been made. Please refer to line 498 in the revised manuscript with track changes.

Point 33: L484 is affected

Response 33: As per the comment, necessary change has been made. Please refer to line 526 in the revised manuscript with track changes.

Point 34: L495 are developing breeds

Response 34: As per the comment, necessary change has been made. Please refer to line 537 in the revised manuscript with track changes.

Point 35: L501 delete 'highly

Response 35: As per the comment, necessary change has been made. Please refer to line 543 in the revised manuscript with track changes.

Point 36: L528 this sentence does not make sense

Response 36: As per the comment, the sentence has been modified to understand better. Please refer to lines 573-574 in the revised manuscript with track changes.

Point 37: L533 up to 2 weeks before antibody protection was reduced.

Response 37: As per the comment, necessary change has been made. Please refer to line 577 in the revised manuscript with track changes.

Point 38: L567 it = the vaccine?

Response 38: Yes, however, the sentence has been removed from the manuscript. Please refer to line 612 in the revised manuscript with track changes.

Point 39: L574 probiotics are not necessarily inactivated.  Generally, they are live non-pathogenic bacteria.

Response 39: As per the comment, the sentence has been modified. Please refer to lines 619-620 in the revised manuscript with track changes.

Point 40: L584 are also shown

Response 40: As per the comment, necessary change has been made. Please refer to line 629 in the revised manuscript with track changes.

Point 41: L597 are SPR02 and DAF6 the phages?  Unclear.

Response 41: Yes, they are phages. As per the comment, necessary change has been made to make it clear. Please refer to line 644 in the revised manuscript with track changes.

Point 42: L615 including those resistant to antibiotics

Response 42: As per the comment, necessary change has been made. Please refer to line 661 in the revised manuscript with track changes.

Point 43: L622 Should the US also be following Denmark and selecting for disease resistance instead of single trait selection on growth rate?  Important to discuss.

Response 42: Thank you for the comment. There is only limited information about the genetic selection based on disease resistance and the available studies are discussed under subsection 5.2. To the best of our knowledge no research in the US related to genetic selection of broiler breeders against APEC infections is available.

Round 2

Reviewer 1 Report

Comments and Suggestions for Authors

I have no comments.